# Workplace Well-Being and Intent to Stay by Health Care Workers Reassigned during the First COVID-19 Wave: Results of a Swiss Survey

**DOI:** 10.3390/ijerph18178976

**Published:** 2021-08-26

**Authors:** Ingrid Gilles, Cédric Mabire, Margaux Perriraz, Isabelle Peytremann-Bridevaux

**Affiliations:** 1Center for Primary Care and Public Health (Unisanté), Department of Epidemiology and Health Systems, University of Lausanne, 1010 Lausanne, Switzerland; Margaux.Perriraz@unisante.ch (M.P.); Isabelle.Peytremann-Bridevaux@unisante.ch (I.P.-B.); 2Institute of Higher Education and Research in Healthcare, Faculty of Biology and Medicine, University of Lausanne, 1010 Lausanne, Switzerland; Cedric.Mabire@chuv.ch

**Keywords:** COVID-19, pandemic, healthcare professionals, reassignment, intent to stay, well-being

## Abstract

Our study aimed at investigating the way not having the choice to be reassigned was associated to a poorer experience of reassignment among health care workers (HCWs) during the first wave of the coronavirus disease 2019 (COVID-19), and indirectly to a lower workplace well-being and reduced intent to stay at the hospital. We also investigated the moderating role of the perceived hospital management responsiveness on these associations. A cross sectional survey was sent to all professionals from 11 hospitals and clinics in the French-speaking part of Switzerland, in July 2020. Out of 2811 professionals who completed the survey, 436 were HCWs reassigned to COVID-19 units during the first wave of the pandemic and constituted our analysis sample. Results indicated that hospital management responsiveness moderated the association between lack of choice and reassignment experience, indicating that the more HCWs perceived responsiveness, the less the lack of choice affected their experience of reassignment and thus their intent to stay and workplace well-being. Lack of choice during reassignments can reduce intent to stay and workplace well-being, in particular if hospital management is not perceived to be responsive during the crisis. Attempts by hospital management to find solutions, such as flexibility in working hours or extraordinary leaves, can alleviate the perceived constraints of reassignment and be considered signs of responsiveness from hospital management.

## 1. Introduction

Coronavirus disease 2019 (COVID-19) became a major public health concern in March 2020 [1]. Constant and unfamiliar changes in health care workers’ (HCWs’) practices and the increased workload during the different waves generated high levels of stress, leading to acute stress disorder and symptoms of depression and anxiety [2,3,4,5]. Moreover, because of the rapid increase in the number of patients with COVID-19, hospitals had to quickly reconfigure workspaces and restructure clinical teams. The frequent restructuring of units and involvement in setting up of specialized pandemic clinics or staging of facility operations exacerbated feelings of uncertainty and stress [6,7]. In this context, many HCWs were reassigned to COVID-19 units, most of the time outside their usual clinical specialty and/or expertise, and additionally had to work extra shifts and longer hours to meet the high volume of patients. Such reassignment can represent an additional source of anxiety for HCWs, who can fear being unable to provide appropriate care if assigned to specialties that do not correspond to their expertise (e.g., non-intensive care unit nurses having to function as intensive care unit nurses) [8].

Actually, reassignments during pandemics are quite unavoidable. In exceptional circumstances, such as the COVID-19 pandemic, duty to care justifies staff redeployment or mobilisation. This duty to care, present in the conduct codes of most health care professions, is based on the fact that health care professionals (i) possess specific irreplaceable skills, (ii) have freely accepted to practice health care professions knowing the associated risks and obligations, and (iii) have a social contract with the community [9]. However, recent SARS and H1N1 outbreaks have highlighted that duty of care may conflict with other duties, such as family duties [10,11]. Up to now, several studies have examined HCWs’ perceptions and attitudes toward duty to care, including in the context of reassignment. In a 2009 study that included 908 professionals, the authors showed that although 60% of respondents found it unethical not to work during a pandemic, 65% would have liked to have the autonomy to decide whether or not to work [12]. In this context, reassignment in frontline units can be perceived by HCWs as extra risk taking [13,14], resulting in lower workplace well-being and willingness to work during pandemics [15], and ultimately leading HCWs to leave their jobs [16,17]. Currently, studies estimate that around 20% of HCWs are thinking of leaving their position due to the COVID-19 pandemic [18,19]. It is thus quite urgent to identify how to conciliate different needs: those of care institutions that must be able to mobilise a large number of professionals in times of heightened need and those of the professionals who have other duties than professional ones and do not want to run significant risks. Understanding how HCWs experienced reassignments in COVID-19 frontline units is central to this purpose.

Several factors, such as control over work or having the choice to work in frontline care units, were shown to play a key role in HCWs’ workplace well-being and intent to stay during pandemics [7]. For example, during the severe acute respiratory syndrome (SARS) outbreak, several studies showed that the question of choice or of autonomy in deciding to work or not was perceived as legitimate by HCWs in some circumstances (e.g., insufficient security, family duty) and did not contradict their duty to work [20]. In fact, HCWs considered lack of choice or coercive decisions imposed by hospital management in high-risk situations to be a failure in moral engagement of hospital management toward its employees [20]. On the opposite, perceiving responsiveness from one’s management (i.e., perceiving that the hospital management is engaged in supporting teams by providing adequate information and security conditions) was described as favouring the development of positive employee–employer relations [21,22]. Same as supervisor support [22], the responsiveness of management could moderate the impact of harsh working conditions on employees’ well-being. However, this has never been explored in the literature [10,23]. In a pandemic context, the fact that hospital management is responsive by providing transparent information and procedures, providing adequate and safe materials, and being reactive in providing clear infection control procedures has been found to decrease the psychological burden of the crisis on HCWs [24,25,26]. Although lack of choice and responsiveness of hospital management were shown to be separately involved in workplace well-being and intent to stay mechanisms, neither the understanding of their conjoint role nor a specific focus on HCWs have been studied so far. To our knowledge, no study currently addresses this gap.

The objective of this study was thus to describe the associations between the experience of reassignment in frontline units and both workplace well-being and intent to stay among HCWs. More precisely, we proposed that the lack of choice in being reassigned was associated to a negative experience of the reassignment and indirectly to lower well-being and decreased intent to stay (mediation). We also proposed that these associations were conditional on how reassigned HCWs perceived hospital management responsiveness: the more hospital management was perceived as responsive towards the crisis, the weaker was the association between lack of choice and reassignment experience (moderated meditation; Figure 1).

## 2. Materials and Methods

### 2.1. Sample and Procedure

The current study is part of a larger project involving all types of professionals working at hospitals during the first wave of COVID-19. In the latter project, professional e-mail addresses of all employees working in 11 hospitals and clinics of the French-speaking part of Switzerland (*n* = 8645) were obtained from the hospitals’ responsible body. Agreements were signed with the hospital responsible body to ensure data security and confidentiality, and communication was made with the professionals by their managements at least 2 days before the beginning of the study. All professionals were then invited to participate in an anonymous electronic survey in July 2020. A reminder was sent 2 weeks after the launch of the survey. Results presented in this article focus on HCWs having indicated that they were reassigned to COVID-19 units during this period. Data from professionals (HCWs or professionals from administration, logistics, etc.) who indicated they were not reassigned were not included. Since information about the number of HCWs reassigned was not available from hospital management, the total number of HCWs actually reassigned cannot be provided.

### 2.2. Measures

The variables included in the survey originated from validated scales or were created specifically with an expert group (research team, members of the hospital medical staff, quality and safety experts, professionals in charge of psychosocial support for HCWs) to match with the context of the pandemic. For the latter, the wording of items was based on a literature search of previous SARS, H5N1, or H1N1 epidemic studies and on interviews conducted with staff physicians, a quality and safety expert, and mediators involved in support services to the teams during the first wave of the pandemic. The reliability of the measured constructs was assessed with principal component analyses and Cronbach’s alphas.

#### 2.2.1. Outcomes

The 2 outcomes of this study were workplace well-being and intent to stay.

Workplace well-being was assessed with the 8 items of the Psychological Well-Being Scale [27] adapted for the workplace by Fisher [28]. The scale assesses self-perceived functioning in areas, such as self-esteem, purpose, and relationships, measured on a 5-point Likert scale. Internal consistency was good (Cronbach’s alpha = 0.88) and a mean score from 1 (poor well-being at work) to 5 (strong well-being at work) was computed.

Intent to stay was measured with a single item that assessed the perceived motivation of professionals to stay after the first wave (“I keep motivated to stay working in my hospital”) on a 5-point Likert scale.

#### 2.2.2. Independent Variables

The following independent variables were measured:

Lack of choice regarding reassignment was measured with a single-item asking if participants had had the choice to accept or decline the reassignment on a 5-point Likert scale ranging from 1 (possibility to choose) to 5 (no choice).

Reassignment experience was measured with 4 items using a 5-point Likert scale (“My colleagues were available when I had questions”; “This experience was humanly enriching”; “I could develop new professional skills”; “This experience was stressful”). Internal consistency was quite good (Cronbach’s alpha = 0.68) and a mean score was computed on the 4 items that ranged from 1 (negative experience of reassignment) to 5 (positive experience of reassignment).

The perceived responsiveness of hospital management was measured with the 5 following items using 5-point Likert scales: “I feel that the crisis was managed effectively within the institution”; “I feel that the management has been responsive to the situation”; “The resources put in place by the management to support the professionals (materials, hotline, information, IT solutions, etc.) met my needs”; “I feel that the management was present and mobilised during the whole period”; “Overall, I felt supported by the hierarchy during the coronavirus first wave”. Internal consistency on the 5 items was good (Cronbach’s alpha = 0.83) and a mean score was computed that ranged from 1 (poor perceived responsiveness) to 5 (strong perceived responsiveness).

Age, gender, length of professional experience, professional groups (dummy variables), the threat of being infected or of infecting others (mean score on four 5-point Likert scale items; Cronbach’s alpha = 0.85), and being involved in intensive care (yes/no) were considered confounding variables.

### 2.3. Statistical Analyses

We treated all variables as continuous variables (except where specified) after checking for distribution linearity and normality. Since the missing value rate was low (<0.5%), we did not estimate missing values and opted for a listwise deletion procedure.

We first conducted descriptive analyses to characterize the data, after which we computed Pearson correlations to assess potential covariations that were due to confounding variables. Finally, as our hypothesis involved both a mediation and a moderation effect, suggesting that the mediation effect depended on the level of our moderator, we used the PROCESS program proposed by Hayes [29]. This program, specifically developed to assess complex models with direct, indirect, and indirect conditional effects, automatically estimates conditional indirect effects (95% bias-corrected CIs estimated with a bias-corrected percentile bootstrap method: 10,000 samples). We chose this method among structural equation modelling because PROCESS allows probing of interactions and provides an index of moderated mediation (the indirect effect of the highest order product with a 95% bootstrapped CI) [30]. Covariates identified with Pearson correlations were entered into the analyses to control for confounding effects.

Descriptive analyses and the PROCESS macro for moderated mediation analyses were performed with SPSS Statistics 26 (IBM Corporation, Armonk, NY, USA); the software program GPower (Heinrich-Heine-Universität Düsseldorf, Düsseldorf, Germany) was used to check that the sample size was adequate for estimation analyses [31].

According to the local ethics committee, the study did not need ethics approval (Req-2020-00695).

## 3. Results

### 3.1. Descriptive and Bivariate Analyses

Of 8645 contacted professionals, 2811 completed the electronic survey (32.5% response rate), and 818 (9.5% of the contacted professionals, 29.1% of the whole respondents’ sample) reported having been reassigned during the first COVID-19 wave. Among the latter, 382 were reassigned to non-COVID-19 units because they had personal vulnerabilities or lived with vulnerable relatives, and 436 were specifically reassigned to COVID-19 units. As the study concerned workplace well-being and intent to stay experienced by HCWs reassigned to frontline units specifically, our analyses concerned the latter 436 respondents (5.0% of the contacted professionals and 15.5% of the whole respondents’ sample). Characteristics of the study sample (*n* = 436), all respondents, and eligible HCWs are reported in Table 1.

Of the 436 reassigned respondents, 76.6 were women; 49.8% of this sample was >40 years of age and 44.8% had been working in the hospital for more than 6 years. Characteristics of the sample respondents were similar to those of the total sample and eligible HCWs except concerning professional groups. Professionals who reported COVID-19 symptoms but did not take a test were more numerous among reassigned respondents than among all respondents.

Concerning the variables in the models, 71.1% of the 436 reassigned HCW respondents stated that they remained motivated to work in their hospitals in the future. Concerning workplace well-being (M = 3.5, SD = 0.8), 28.1% of the 436 respondents expressed a high level of well-being (4.0 to 5.0 on a 5-point scale), 50.8% reported a medium level (between 3.0 and 3.9 on a 5-point scale), and 21.1% expressed a low level (between 1.0 and 2.9 on a 5-point scale). Concerning lack of choice (M = 3.2, SD = 1.4), 32.3% of the 436 respondents answered they had choice (4.0 to 5.0 on a 5-point scale), 19.5% answered they had a limited choice (between 3.0 and 3.9 on a 5-point scale), and 48.2% answered they had no choice (between 1.0 and 2.9 on a 5-point scale). Concerning reassignment experience (M = 3.4, SD = 0.8), 25.6% of the 436 respondents reported a good experience with reassignment (4.0 to 5.0 on a 5-point scale), 45.3% reported a neutral experience (between 3.0 and 3.9 on a 5-point scale), and 29.1% reported a negative experience (between 1.0 and 2.9 on a 5-point scale). Concerning the perceived responsiveness of hospital management in front of the pandemic crisis (M = 3.7, SD = 0.8), 36.0% of the 436 respondents felt that management had been responsive during the first wave of COVID-19 (4.0 to 5.0 on a 5-point scale), 40.1% felt it had been moderately responsive (between 3.0 and 3.9 on a 5-point scale), and 23.9% thought it had not been responsive (between 1.0 and 2.9 on a 5-point scale).

Bivariate correlations (Table 2) showed weak to moderate associations between the variables in the moderated mediation model and the confounding variables (r ranging from |0.00| to |0.21|). Confounding variables were therefore not entered as covariates in analyses. Correlations between predicting variables ranged from 0.22 to 0.45, indicating no multicollinearity issues.

### 3.2. Moderated Mediation Analyses

We expected trust in hospital management of the COVID-19 crisis to moderate the indirect path between perceived lack of choice and both intent to stay and workplace well-being through the reassignment experience. The two moderated mediation analyses showed a significant overall index of moderated mediation for both outcomes (intent to stay: index 0.03, 95% bootstrapped CI, 0.01 to 0.05; workplace well-being: index 0.04, 95% bootstrapped CI, 0.01 to 0.06), suggesting a moderated mediation effect on both outcomes.

The results reported in Figure 2 (see details in Appendix A, Table A1) reveal that lack of choice was significantly associated with reassignment experience: the more HCWs perceived a lack of choice, the less positive the experience of reassignment was. Moreover, reassignment experience was significantly associated with both intent to stay and workplace well-being: a negative experience was associated with less intent to stay and lower workplace well-being. In addition, the direct effect of lack of choice, after we controlled for reassignment experience, was no longer significant for intent to stay or for workplace well-being, suggesting a mediation effect for intent to stay and a partial mediation effect for workplace well-being. According to these mediation effects, the more professionals felt there was a lack of choice concerning the reassignment, the more negative their experience of the reassignment was, which decreased their intent to stay in the hospital and their workplace well-being.

Finally, as hypothesized, we observed that the indirect effects of lack of choice on intent to stay and workplace well-being (through reassignment experience) were conditional on the perceived responsiveness of hospital management. In fact, the more that HCWs perceived their hospital management to be responsive, the less that lack of choice affected their experience of reassignment (Figure 3) and thus their intent to stay and workplace well-being.

## 4. Discussion

In the current COVID-19 pandemic context, our results suggest that both having the choice to be reassigned and perceived responsiveness of hospital management can increase intent to stay and workplace well-being. Despite the fact that many studies have already reported the negative consequences of the pandemic situation on the physical and mental health of frontline HCWs, including reassigned HCWs [32,33,34,35], few have investigated the impact of these harsh working conditions on intent to leave. In fact, intent to leave or unwillingness to work in pandemic situations has already been reported, starting with the 1918 influenza pandemic and up to the more recent 2013 SARS outbreak [16,36,37]. Infectious disease for which treatment or vaccination is limited, such was the case for COVID-19 during the first wave, has been associated with very low intent to stay rates indeed [38,39].

The association found in our study between having the choice to be reassigned and well-being or intent to stay could be explained by the fact that having the choice can provide HCWs with a sense of control over events. Actually, control over work has been described as a strong determinant of nurses’ retention [40] and as a major coping factor associated with psychological distress [41] and job satisfaction [42] among HCWs during pandemics. The particular uncertainty associated with the COVID-19 pandemic surely increased the need among HCWs to control their working environment [43]; we can imagine this need to have been stronger among reassigned HCWs. In this sense, allowing HCWs either to choose to be reassigned or to choose the period during which they would be reassigned could help them gain a sense of control during this uncertain situation.

This is coherent with research suggesting that reassignment should be organized in the form of a contract between the hospital and the HCWs rather than in the form of a coercive one-way decision [20]. The autonomy in choosing to work or not therefore appears to be an important aspect, especially when there are tensions between family and professional responsibilities [20].

Beyond the ethical issue raised by the duty of care in times of a pandemic, our study suggests that lack of choice, or, conversely, imposed reassignments, may be associated with reduced professional well-being and a willingness to leave the profession. This could be because HCWs’ perceptions of duty to care have evolved in recent outbreaks [20,44], showing that legitimate limitations to HCWs’ duty to care exist [20]. Whereas these limitations mainly depend on HCWs’ personal constraints or responsibilities (e.g., living with a relative vulnerable to the disease), they also depend on hospital management: HCWs have a moral duty to work despite the risks that they are exposed to, but hospital management is obliged to minimize the risks of its employees [20]. If HCWs feel that the hospital does not provide them with enough security and support in such contexts, this may represent a legitimate limitation to the duty to care. In fact, several authors have proposed that HCWs view duty of care not as a one-way commitment but as a reciprocal one [20,45,46]. In our study, the perception of the responsiveness of hospital management could be interpreted as an indirect appraisal of this reciprocity: by its perceived responsiveness or non-responsiveness in the face of a crisis, hospital management signals to HCWs whether or not it respects its own duty of care. Ives and colleagues [47] propose that communication and visibility of hospital reactiveness is necessary “to encourage the feeling that the needs of workers are being acknowledged” and thus participate in reciprocity. In addition, perceived organizational reaction and responsiveness have been found to help HCWs’ ability to cope with the current pandemic situation [24]. Thus, being reactive during critical situations and making this visible to employees represents a crucial approach for hospital management to build a reciprocal relationship, in particular with its frontline employees.

An interesting result in our study concerns the mediating role of reassignment experience between lack of choice and both workplace well-being and intent to stay. In fact, perception of control over work, autonomy, or lack of choice are usually found to be directly associated with well-being, as proposed in the job-demand-control model [48] or with intent to stay [49]. Our results show that this association is mediated by HCWs’ experience. Actually, this mediating role of reassignment experience is not surprising as experience can be seen as an overall perception of the situation related to job satisfaction [50], one of the strongest predictors of HCWs’ intent to stay [51]. Several studies have pointed out the mediating role of job satisfaction between working conditions and job outcomes [52,53]. According to them, employees stay in their position not because they have actual adequate working conditions but because these working conditions are appraised positively or generate satisfaction. We could apply the same reasoning in our case: intent to stay or well-being would not be due to the lack of choice per se, but to the fact that lack of choice generates a negative appraisal of the situation.

The main strengths of this study are that it involved investigating an understudied topic and that it included diverse reassigned HCWs (including nurses, physicians, physiotherapists, etc.) in 11 Swiss hospitals. However, the following two limitations need to be considered. First, the response rate may be considered low. Such response rates are, however, close to those from similar studies, especially in a pandemic context, which may have had an impact on participation in a survey. In addition, the characteristics of our respondent sample and those of the eligible professionals were almost identical. A second limitation is the one-time cross-sectional measurement of well-being at work. Indeed, as the current pandemic period is characterized by recurrent waves and uncertainties, several measurements would allow fluctuations in well-being to be captured. Moreover, a longitudinal design could help confirm causality. The current acute and highly worrisome pandemic context nonetheless prevents oversolicitation of professionals and makes the performance of a longitudinal study unrealistic.

## 5. Conclusions

Our study highlights that, for HCWs reassigned during the first wave of COVID-19 to unfamiliar work environments under demanding conditions, having the choice to accept or decline the reassignment may have an indirect impact on their well-being and intent to stay. However, this can be compensated by the fact that hospital management is perceived as responsive during crises.

This result has concrete implications for hospital administration, who should consider reassignments not as an obligation imposed to HCWs but as a contract of reciprocity between hospital management and HCWs. In addition, lack of choice is mainly relevant when HCWs perceive barriers to working during pandemics (e.g., childcare, risk to self). Understanding these barriers and proposing solutions for overcoming them, such as flexible working hours or agreements about extraordinary leave, for example, can be concrete signs of responsiveness that hospital management can provide to HCWs. Considering these solutions, which need to be anticipated in preparedness planning, is crucial to ensure engagement of workforces during demanding periods.

## Figures and Tables

**Figure 1 ijerph-18-08976-f001:**
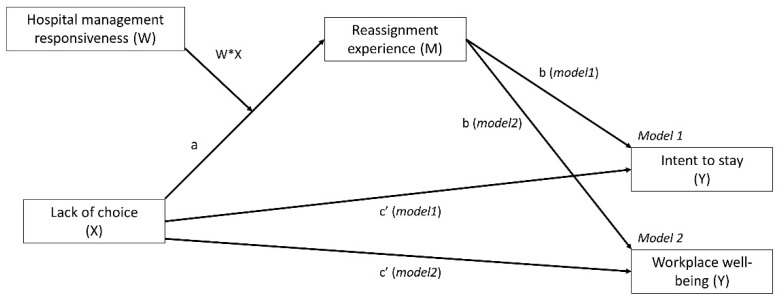
Moderated meditation models for intent to stay (model 1) and workplace well-being (model 2). Note: X is the predicting variable, Y is the outcome variable, M is the mediator, W is the moderator, and W * X is the interaction between the predicting variable and the moderator; a, b, and c’ are paths in the model.

**Figure 2 ijerph-18-08976-f002:**
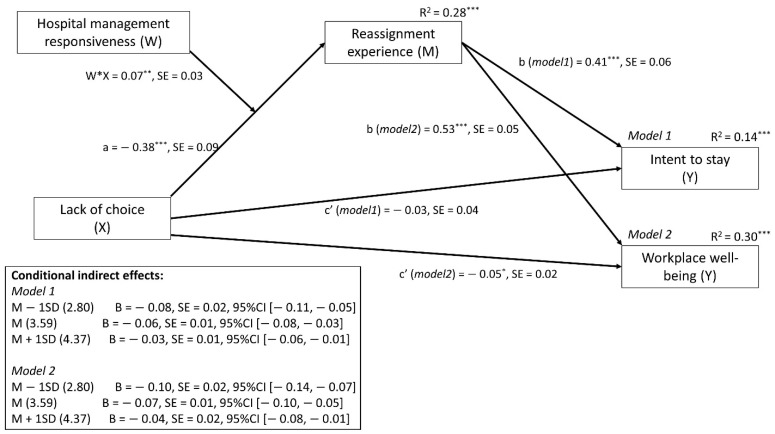
Final model showing moderated mediation. Effect of lack of choice on intent to stay (model 1) and well-being (model 2) is mediated by reassignment experience, and this effect is conditioned (moderated) by hospital management responsiveness. Note: *** *p* < 0.001; ** *p* < 0.01; * *p* < 0.05.

**Figure 3 ijerph-18-08976-f003:**
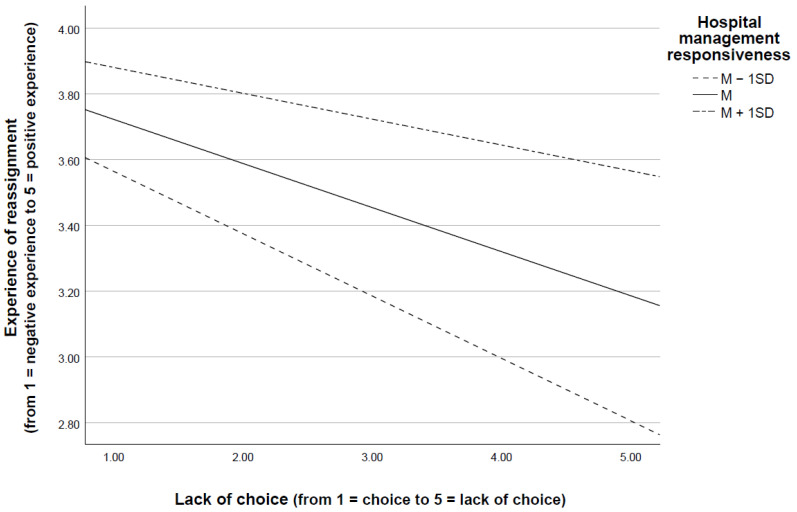
Lack of choice and experience of reassignment at different levels (M and M ± 1 SD) of hospital management responsiveness. Note: M = Mean; M + 1 SD = Mean plus one standard deviation; M − 1 SD = Mean minus one standard deviation.

**Table 1 ijerph-18-08976-t001:** Characteristics of the reassigned respondents (study sample A), all survey respondents (B), and eligible health care workers (HCWs) (C).

Characteristics	A *n* (%)	B *n* (%)	C *n* (%)
Count.	436	2811	8645
Gender			
Women	327 (76.6)	2129 (78.1)	6510 (75.3)
Men	100 (3.4)	598 (21.9)	2135 (24.7)
Age			
<30 years	95 (22.1)	507 (18.5)	1729 (20.0)
30–39 years	119 (27.7)	739 (27.0)	2282 (26.4)
40–49 years	125 (29.1)	687 (25.0)	2058 (23.8)
≥50 years	91 (21.2)	807 (29.5)	2576 (29.8)
Length of professional experience			
<3 years	96 (22.3)	659 (24.1)	3130 (36.2)
3–5 years	97 (22.5)	554 (20.2)	1565 (18.1)
6–10 years	80 (18.6)	557 (20.4)	1279 (14.8)
>10 years	158 (36.7)	967 (35.3)	2671 (30.9)
Professional group			
Administration (patients’ reception staff) and logistics	34 (7.9)	702 (24.9)	2447 (28.3)
Physicians	36 (8.4)	191 (6.9)	1020 (11.8)
Nurses and nurse assistants	273 (63.9)	1323 (47.1)	3882 (44.9)
Other HCWs (e.g., psychologists, physiotherapists, radiologists, pharmacists)	81 (19.0)	463 (16.4)	752 (8.7)
Other (e.g., students)	12 (2.8)	132 (4.7)	545 (6.3)
Reassignments			
No reassignment	NA	1960 (70.6)	--
Reassignment in a facility not dedicated to COVID-19 patient care ^1^	NA	382 (13.7)	--
Reassignment to COVID-19 patient care facilities	NA	436 (15.7)	--
Having had symptoms compatible with COVID-19			
Yes and confirmed by a positive test	16 (3.7)	113 (4.1)	--
Yes but not confirmed by a test	59 (13.5)	258 (9.3)	--
Yes but the test was negative	56 (12.8)	284 (10.2)	
No or don’t know	305 (70.0)	2122 (76.4)	--

Note: ^1^ For example, HCWs who were reassigned because they were personally at risk for COVID-19; -- = data not available from eligible professionals.

**Table 2 ijerph-18-08976-t002:** Correlations (Pearson for continuous variables and Spearman for the ordinal variable) among the main study variables and confounding variables.

	2	3	4	5	6	7	8	9
1. Gender	0.00	−0.04	0.00	0.02	−0.04	−0.05	0.00	0.01
2. Age	--	0.60 ***	−0.09	0.07	−0.07	0.05	0.04	0.02
3. Length work. exp.		--	−0.08	0.02	−0.02	0.05	0.02	−0.05
4. Threat of infection			--	0.18 **	−0.18 **	−0.21 **	−0.08	−0.09
5. Lack of choice				--	−0.35 ***	−0.22 ***	−0.28 ***	−0.19 ***
6. Reass. experience					--	0.45 ***	0.54 ***	0.32 ***
7. Hosp. man. respons.						--	0.53 ***	0.49 ***
8. WWB							--	0.49 ***
9. Intent to stay								--

Note: Length work. exp. = length of working experience; Reass. experience = reassignment experience; Hosp. man. respons. = hospital management responsiveness; WWB = workplace well-being; *** *p* < 0.001; ** *p* < 0.01

## Data Availability

The data that support the findings of this study are available upon reasonable request.

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
