# Peer review of "Workplace Well-Being and Intent to Stay by Health Care Workers Reassigned during the First COVID-19 Wave: Results of a Swiss Survey"

_ijerph, 2021, doi:10.3390/ijerph18178976_

Round 1

Reviewer 1 Report

The spread of the COVID-19 pandemic, which came without warning, are providing many difficult conditions. In particular, hospital health care workers (HCWs) would be anxious at the frontline units, and conflicts between hospitals and employees are inevitably amplified in a hospital environment where collaboration does not take place. The purpose of this study is to identify the relationship between intent to stay and workplace well-being according to the lack of choice in the front-line department of HCWs, and to verify the role of hospital management responsiveness. The research result was suggested a human resource management strategy method for HCWs in COVID-19 conditions. The study's conclusions represent a significant contribution.

Nevertheless, we would like to offer some additional suggestions as follows.

  1. Introduction

- Please state the importance of this study more clearly.

- Please suggest the difference from this study by comparing it with the previous study.

- This study is a causal relation model analysis study through hypothesis setting. They should clearly present results of the prior research before setting up a hypothesis.

  1. Materials and Methods

- Please provide specific measurement items for reassignment experience, intent to stay, and hospital management responsiveness.

- Please suggest a method for verifying the validity and causal relation analysis of the measurement items.

  1. Results

- I also suggest to articulate the study period to see how up-to-date this study is.

  1. Conclusion

- Please write the theoretical and academic implications separately.

- It is necessary to present the distinction between this study and previous research results. For example, lack of choice has no direct effect on intent to stay.

Appendix A

Table 1 is not sufficiently clear. It is not clear enough which data correspond (model 1, 2 or hypothesized) to which relation. Authors need to edit the table 1 to improve clarity.

Thanks.

Reviewer 2 Report

Dear Authors,

I recommend the paper for publication after some revisions, namely:

  • Methodology should be better presented in the Abstract.
  • In the “Discussion” section you refer to additional literature, not mentioned in the Introduction. I suggest presenting the theoretical background/relevant studies before you start the empirical part of your paper.
  • The Literature Review presented in the Introduction section seems to be limited and have a limited number of current research in the field.
  • Rethink the “Conclusions” section - at the moment it lacks the Limitations and the Future Research subsections. Try to:
    1. refer to the paper aim,
    2. highlight key findings in your “Results” section,
    3. place the paper within the context of how your research advances past research about the topic,
    4. describe how a previously identified gap in the literature (your literature review section) has been filled by your research,
    5. demonstrate the importance of your ideas and recommendations/suggestions,
    6. define the limitations of your research,
    7. propose possible new or expanded ways of thinking about the research problem.
  • The referencing style must be adopted to the Journal one.

I do hope you find the comments helpful as you move forward with your conference paper.

Reviewer 3 Report

Suggest that the abstract is put more into a research format to make it clear.

Concerned that this should have had some ethical review. Usual would be given a reference number to support this claim.

2.1 not clear - said all healthcare workers then stated that focusing only on those that were reassigned. This needs to be presented in more detail to explain this.

2.2 not clear where the questions used in the survey came from with the use of 'or'. Did the questions come from all of these or just some selected as appropriate. Needs to be explained more to make it clear.

Need to explain how the participants were recruited.

3. results are not clearly presented. Suggest put in percentage of the total population.

Line 159 - rather concerning that a number of participants who had COVID symptoms did not get a test. This needs to have a number and percentage against it so that it is clear how many this refers to. This is not clearly discussed either and needs to be.

Line 162 is this just about the 436 who were reassigned - not clear.

line 166 - quite responsive to what - need to explain.

Line 252, 256 - think this should be duty of care

Line 282 is not clear and needs rewriting.

4. Discussion second paragraph is too long and should be divided more.

One of the key aspects that are alluded to here but not discussed is the fact that these healthcare professionals had control over their work environment which is discussed in the literature as being significant to staff satisfaction and intent to stay. Suggest that this be added to this discussion.
